

# Hydrogen peroxide induced loss of heterozygosity correlates with replicative lifespan and mitotic asymmetry in *Saccharomyces cerevisiae*

Emine Güven[1,2,*], Lindsay A. Parnell[1,3,*], Erin D. Jackson[1], Meighan C. Parker[1], Nilin Gupta[1], Jenny Rodrigues[1] and Hong Qin[1,4]

[1] Department of Biology, Spelman College, Atlanta, Georgia, United States
[2] Current affiliation: Department of Computer Science and Engineering, University of Tennessee at Chattanooga, Chattanooga, Tennessee, United States
[3] Current affiliation: Program of Molecular Genetics and Genomics, Division of Biology and Biomedical Sciences, Washington University in St. Louis, St. Louis, Missouri, United States
[4] Current affiliation: Department of Computer Science and Engineering, Department of Biology, Geology, and Environmental Science, SimCenter, University of Tennessee at Chattanooga, Chattanooga, Tennessee, United States
* These authors contributed equally to this work.

Corresponding author
Hong Qin, hong-qin@utc.edu

## ABSTRACT

Cellular aging in *Saccharomyces cerevisiae* can lead to genomic instability and impaired mitotic asymmetry. To investigate the role of oxidative stress in cellular aging, we examined the effect of exogenous hydrogen peroxide on genomic instability and mitotic asymmetry in a collection of yeast strains with diverse backgrounds. We treated yeast cells with hydrogen peroxide and monitored the changes of viability and the frequencies of loss of heterozygosity (LOH) in response to hydrogen peroxide doses. The mid-transition points of viability and LOH were quantified using sigmoid mathematical functions. We found that the increase of hydrogen peroxide dependent genomic instability often occurs before a drop in viability. We previously observed that elevation of genomic instability generally lags behind the drop in viability during chronological aging. Hence, onset of genomic instability induced by exogenous hydrogen peroxide treatment is opposite to that induced by endogenous oxidative stress during chronological aging, with regards to the midpoint of viability. This contrast argues that the effect of endogenous oxidative stress on genome integrity is well suppressed up to the dying-off phase during chronological aging. We found that the leadoff of exogenous hydrogen peroxide induced genomic instability to viability significantly correlated with replicative lifespan (RLS), indicating that yeast cells' ability to counter oxidative stress contributes to their replicative longevity. Surprisingly, this leadoff is positively correlated with an inverse measure of endogenous mitotic asymmetry, indicating a trade-off between mitotic asymmetry and cell's ability to fend off hydrogen peroxide induced oxidative stress. Overall, our results demonstrate strong associations of oxidative stress to genomic instability and mitotic asymmetry at the population level of budding yeast.

## INTRODUCTION

The budding yeast *Saccharomyces cerevisiae* is a model for cellular aging (*Kaeberlein, 2010*; *Ludovico et al., 2012*). Yeast aging can be studied by replicative aging and chronological aging based on dividing and non-dividing cells, respectively (*Fabrizio & Longo, 2003*; *Longo et al., 2012*). Replicative lifespan (RLS) refers to the number of times a cell undergoes the cell cycle (*Defossez, Park & Guarente, 1998*; *Qin & Lu, 2006*; *Wei et al., 2008*). Chronological lifespan (CLS) measures the amount of time it takes for cells to lose their proliferation potential in stationary phase (*Fabrizio & Longo, 2003*; *Qin & Lu, 2006*; *Qin, Lu & Goldfarb, 2008*). Both replicative aging and chronological aging of yeast cells lead to increased genomic instability, as demonstrated by the age-dependent increase of loss of heterozygosity (LOH) (*McMurray & Gottschling, 2003*; *McMurray & Gottschling, 2004*; *Qin, Lu & Goldfarb, 2008*). As a cellular organism, it is in the best interest for yeast mother cells to keep oxidatively damaged molecules to themselves, rather than pass them down to daughter cells, a phenomenon known as mitotic asymmetry (*Henderson, Hughes & Gottschling, 2014*; *Higuchi-Sanabria et al., 2014*; *Hill et al., 2016*; *Jazwinski, 1990*; *Kennedy, Austriaco & Guarente, 1994*; *Lai et al., 2002*; *McFaline-Figueroa et al., 2011*; *Thayer et al., 2014*; *Yang et al., 2015*). Mitotic asymmetry can break down in both replicatively and chronologically aged cells, and lead to high levels of genomic instability in daughter cells (*Henderson & Gottschling, 2008*; *McMurray & Gottschling, 2003*; *McMurray & Gottschling, 2004*; *Qin, Lu & Goldfarb, 2008*).

The free radical theory of aging argues that biological systems age because free radicals react with macromolecules and disturb key pathways that are vital to maintaining the overall functional and genomic integrity of cells (*Harman, 1956*; *Ristow & Schmeisser, 2011*; *Yu et al., 2012*). Reactive oxygen species (ROS), a form of free radicals, are natural by-products of the mitochondrial respiratory chain (*Cheeseman & Slater, 1993*). The endogenous levels of ROS, such as superoxide anions and hydrogen peroxide, are often low and play important roles in signaling transduction, defense, and other normal cell functions (*Blagosklonny, 2008*; *Rahman, 2007*; *Weinberger et al., 2010*). At low levels, hydrogen peroxide can promote yeast chronological longevity (*Mesquita et al., 2010*), a phenomenon known as hormesis (*Belz & Piepho, 2012*; *Blagosklonny, 2008*; *Blagosklonny, 2009*; *Boxenbaum, 1991*; *Ludovico & Burhans, 2014*; *Martins, Titorenko & English, 2014*; *Ristow & Schmeisser, 2011*; *Weinberger et al., 2007*). At high levels, superoxide, hydrogen peroxide, and singlet oxygen can oxidize lipids, proteins, and nucleic acids (*Moradas-Ferreira et al., 1996*). The impaired cellular structures and functions induced by damaged molecules can result in altered metabolic homeostasis and abnormal cell growth, which has been argued to be a cause of aging (*Harman, 1956*; *Ristow & Schmeisser, 2011*).

Substantial evidence support the role of ROS in yeast replicative aging and asymmetric age partition between mother and daughter cells (*Higuchi-Sanabria et al., 2014*). Replicatively older cells showed higher levels of superoxide anion and hydrogen peroxide than younger cells, as measured by dihydroethidium and dihydrorhodamine, respectively (*Lam et al., 2011*; *Scheckhuber et al., 2007*). Mitochondria in mother cells were found to gradually lose their inner membrane potential and lead to an iron-sulfur cluster defect

and nuclear genome instability (*Veatch et al., 2009*). This mitochondria dysfunction can lead to increased levels of ROS (*Breitenbach et al., 2014*). Mother cells tend to retain mitochondria with higher superoxide levels than those in daughter cells (*McFaline-Figueroa et al., 2011*). The asymmetric partition of mitochondria between mother and daughter cells can be enhanced by increased retrograde actin cable flow (*Higuchi et al., 2013*). Consistent with the role of ROS, decreased release of mitochondrial reactive oxygen can extend RLS (*Barros et al., 2004*), and supplementation of spermine/spermidine, free-radical scavengers, can increase yeast RLS (*Eisenberg et al., 2009*).

The roles of superoxide anion and hydrogen peroxide in yeast chronological aging have also been investigated. Calorie restriction is found to extend chorological lifespan through moderate increase of hydrogen peroxide that leads to activation of superoxide dismutase and inhibition of superoxide anions (*Mesquita et al., 2010*; *Weinberger et al., 2010*). Heavy water can suppress endogenous ROS and extend yeast CLS (*Li & Synder, 2016*).

We previously found that yeast natural isolates tend to hold off the age-dependent increase of LOH during chronological aging, as shown by the late-onset of LOH increase when compared to the drop of viability (*Qin, Lu & Goldfarb, 2008*), and this ability is significantly associated with yeast RLS and a measure of mitotic asymmetry (*Qin, Lu & Goldfarb, 2008*). The CLS dependent change of LOH was independently observed by another research group (*Maxwell, Burhans & Curcio, 2011*). Age-dependent genomic instability has been attributed to oxidative damage to chromosomal DNA (*Burhans & Weinberger, 2007*; *Burhans & Weinberger, 2012*; *McMurray & Gottschling, 2004*; *Weinberger et al., 2007*).

In the present study, we seek to test whether there is a detectable statistical link between yeast lifespan variations and oxidative stress induced genomic instability at the population level.

## MATERIALS AND METHODS

### Yeast strains and $H_2O_2$ treatment

Strains with $MET15^{+/-}$ have been described previously (*Qin, Lu & Goldfarb, 2008*). Yeast cells were grown overnight at 30 °C in Yeast extract Peptone Dextrose (YPD) media. On the next day, yeast cultures were diluted to $OD_{600}$ of 0.6 in fresh YPD and grown for an additional two hours to ensure that the optical density reached between 0.8 and 0.9, the mid-log phase. Cells were then harvested, washed with water to remove YPD, and sonicated for 4 min using a Fisher Scientific FS20D water-bath sonicator to disperse cell clumps.

Treatment of yeast cells with $H_2O_2$ was modified from other studies (*Diezmann & Dietrich, 2009*; *Yu et al., 2012*). Reactions were carried out by mixing 4 μl of yeast cells, 16 μl of water, and 20 μl of the 2-fold $H_2O_2$ working stocks. Tubes of cells in $H_2O_2$ treatment were incubated on a nutator for 3 h at 30 °C. For most strains, the final concentrations of $H_2O_2$ for cell treatment were typically in a group of ten: 0, 0.005, 0.01, 0.025, 0.05, 0.075, 0.1, 0.15, 0.2, and 0.3%. For strains that are more sensitive or more tolerant to $H_2O_2$, this concentration series was adjusted lower or higher, respectively. At the end of incubation, all reaction tubes were filled with 960 μl of water

(a 50 fold dilution) and chilled on ice for 5 min. Cells were sonicated in a FS20D water-bath sonicator for 2 min. We spread 250 µl of each reaction mix onto 150 mm plates with the Met15 Lead Assay (MLA) media (*Qin, Lu & Goldfarb, 2008*). Plates were spread in triplicates for each $H_2O_2$ concentration. Plates were incubated at 30 °C for 2 days. Some plates were also left at room temperature for additional days for colony colors to develop a better contrast for LOH scoring. Colonies were visually assessed for color-sectoring patterns and tallied using a Bantex Colony Counter. We tallied the number of white, black, half-black, quarter-black, three-quarter black colonies and others. We ignored color-sector patterns that were less than one-eighth. For each experiment, viability at conditions with only water was deemed as 100%. Viability at a specific concentration of $H_2O_2$ was estimated from the average number of colony forming units (CFUs) at this concentration divided by the average number of CFUs at conditions with only water. Strains were assayed 2–5 times, for a total of 35 successful experiments performed with the 12 yeast strains under study.

## Data and analysis codes

Fitting and analysis were conducted in the R statistical environment. Sample codes and data can be found on GitHub (https://github.com/hongqin/H2O2LOH_PeerJ2016). An R markdown script, "summary_analysis_LOHH2O2.Rmd," is provided in the GitHub repository to reproduce key analyses and figures. A summary of the results in EXCEL format is provided in Table S1. The ratios of $C_b/C_v$ were directly estimated from each experiment and then averaged for each strain. Data for replicative and CLS, and mitotic asymmetry measures were obtained from our previous studies (*Qin & Lu, 2006*; *Qin, Lu & Goldfarb, 2008*), and are also available in the same GitHub repository. Average replicative lifespan (ARLS) indicates the average RLS for each strain that was previously measured using micro-dissections (*Qin & Lu, 2006*). CLS in the dataset indicates average CLS for each strain that was previously measured using colony-forming units (*Qin & Lu, 2006*).

# RESULTS

## Overview of the experiment

To better understand the interconnection between oxidative stress, genomic instability, mitotic asymmetry, and CLS in *S. cerevisiae*, we used exogenous $H_2O_2$ to induce an oxidative stress response in yeast cells and quantified their $H_2O_2$ dose-response curves of LOH and viability. LOH is an effective method for gauging genome integrity in yeast cells (*McMurray & Gottschling, 2003*; *McMurray & Gottschling, 2004*). We previously introduced heterozygosity on the *MET15* locus ($MET15^{+/-}$) by knocking out one copy of the wild-type allele using a kanamycin-resistance marker (*Qin, Lu & Goldfarb, 2008*). LOH can be monitored when $MET15^{+/-}$ strains are converted into homozygous $MET15^{-/-}$ strains that can generate black colors on lead-containing medium (Fig. 1). In other words, we can observe half of the LOH events at the $MET15^{+/-}$ locus.

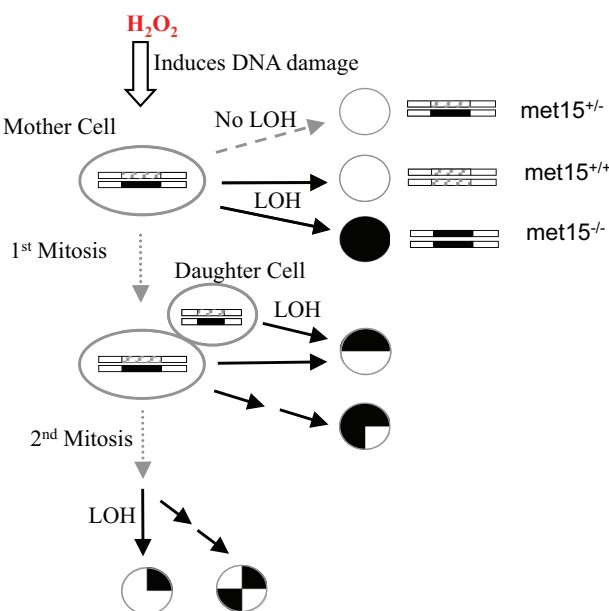

**Figure 1 Experimental scheme to measure the exogenous hydrogen peroxide effect on genome instability at the *MET15* locus.** Cells were treated with various doses of exogenous hydrogen peroxide, and then transferred to plates of lead-containing media.

## Quantitative measurement of $H_2O_2$–dose dependent changes of viability and LOH

The exogenous $H_2O_2$ dose-dependent changes of viability, black colonies, and half-black colonies are generally sigmoid in the studied yeast strains (Figs. 2 and S1). Half-black colonies indicated LOH occurred after cells divided on MLA plates (Fig. 1).

As shown in Fig. 2, we quantified viability $s$ as a function of hydrogen peroxide concentration, $[H_2O_2]$, using a sigmoid mathematical function, similar to our previous study (*Qin, Lu & Goldfarb, 2008*):

$$s([H_2O_2]) = \frac{1}{1 + \left(\frac{[H_2O_2]}{C_v}\right)^{w_v}},$$

where $C_v$ is the $H_2O_2$ concentration at the midpoint of viability, and $w_v$ is a weight parameter.

We quantified the percentage of black colonies, $b([H_2O_2])$, as a function of $[H_2O_2]$:

$$b([H_2O_2]) = b_{max} - \frac{b_{max} - b_{min}}{1 + \left(\frac{[H_2O_2]}{C_b}\right)^{w_b}}$$

where $C_b$ is the $H_2O_2$ concentration at the midpoint of black colony change, $b_{max}$ and $b_{min}$ is the maximum and minimum of $b([H_2O_2])$, and $w_b$ is a weight parameter. Likewise, the change of half-black colonies was quantified:

$$b_{0.5}([H_2O_2]) = b_{0.5\,max} - \frac{b_{0.5\,max} - b_{0.5\,min}}{1 + \left(\frac{[H_2O_2]}{C_{b0.5}}\right)^{w_{b0.5}}}$$

where $C_{b0.5}$ is the $H_2O_2$ concentration at the midpoint of half-black colony change.
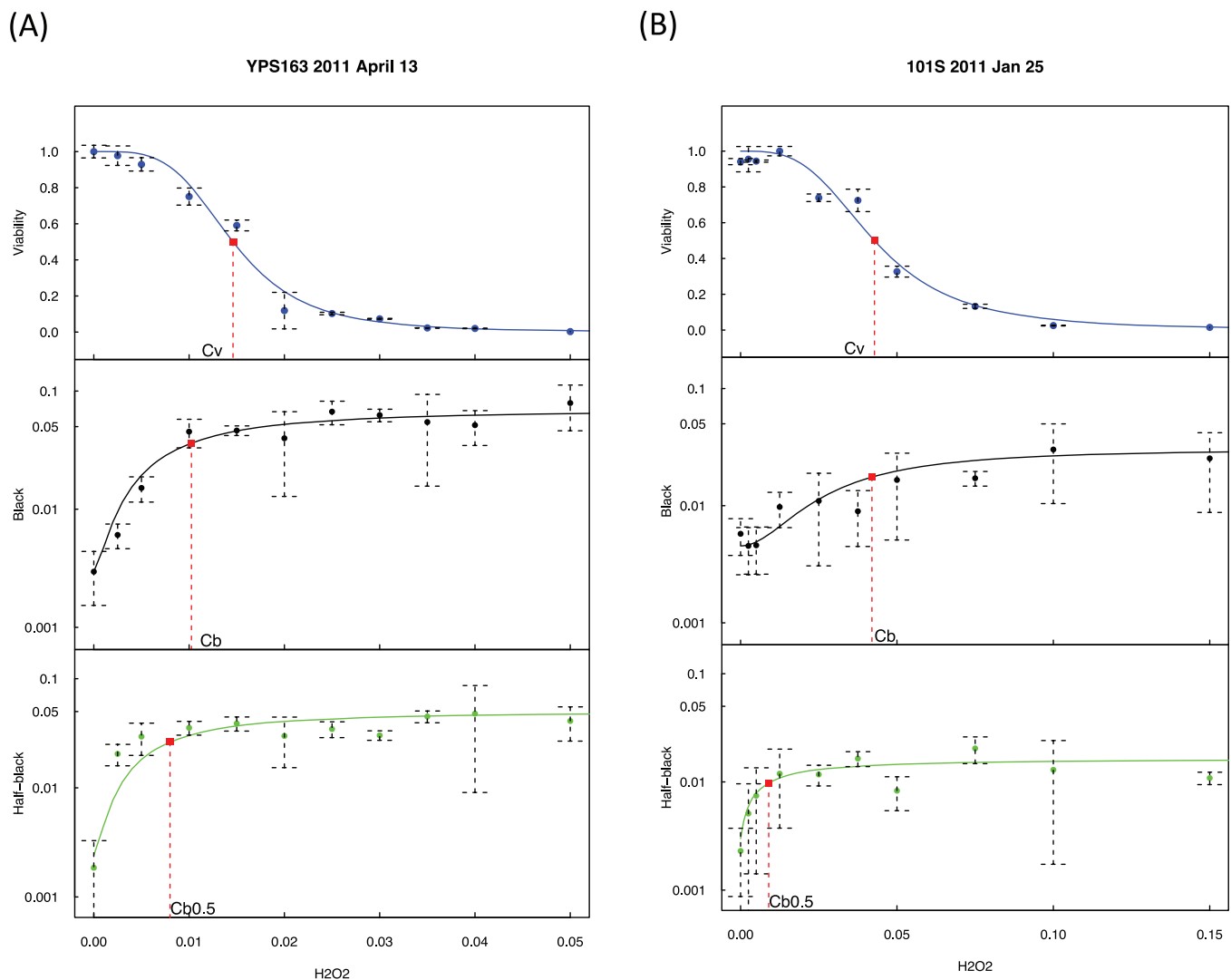

**Figure 2** **Exogenous hydrogen peroxide induces dose-dependent increase of loss of heterozygosity and drop of viability.** Experimental results of two strains are presented, with strain YPS163 in (A) and strain 101S in (B). The vertical axes indicate the fractions of viabilities, black colonies, and half-black colonies. The concentrations of hydrogen peroxide in the horizontal axes are indicated in percentages estimated by volumes. Dashed bars indicate standard deviations. Red squares indicate the mid-transition points.

We detected a strong association between $C_v$ and $C_b$ with $R^2 = 0.77$ and p-value = 0.00019 using the averaged values of each strain (Fig. S1), indicating that hydrogen peroxide induced LOH and loss of viability co-vary, demonstrating the consistency of our approach.

## Genomic tolerance to oxidative stress correlates with replicative lifespan

$C_b$ is lower than $C_v$ in seven out of the 12 strains under study (Figs. 2 and S2; Table S1). The median and mean values of $C_b/C_v$ are 0.60 and 0.92, respectively, indicating that lower doses of $H_2O_2$ are needed to trigger genomic instability than to trigger the drop of viability. In other words, genome integrity is more sensitive to oxidative stress

than cellular viability during exogenous $H_2O_2$ treatment. Strains with higher $C_b/C_v$ values, such as YPS128 in Fig. 3A, indicates their genome integrity is more tolerant of hydrogen peroxide. Strains with lower $C_b/C_v$ values, such as M32 in Fig. 3A, indicate their genome integrity is more sensitive to hydrogen peroxide. Linear regression analysis reveals that average RLS, ARLS, is positively correlated with $C_b/C_v$ with p-value = 0.039 and $R^2$ = 0.36, suggesting that genomic tolerance to oxidative stress is associated with replicative longevity.

## Trade-off between genomic tolerance to oxidative stress and mitotic asymmetry in mother cells

We previously measured an inverse proxy for the mitotic asymmetry, $L_0$, of these yeast strains during normal growth conditions (*Qin, Lu & Goldfarb, 2008*). This inverse proxy for mitotic asymmetry, $L_0$, is the ratio of daughter-cell LOH versus mother-cell LOH in YPD. Small $L_0$ values suggest mother cells can more effectively keep genotoxic stress factors to themselves and thereby provide their daughter cells with healthier lives (*Qin, Lu & Goldfarb, 2008*). Regression analysis revealed that $C_b/C_v$ significantly correlates with $L_0$, with p-value = 0.0079 and $R^2$ = 0.56 (Fig. 3B). Based on the positive correlation between $C_b/C_v$ and $L_0$, yeast cells with better mitotic asymmetry, such as M1–2 and M13, are more sensitive to genotoxic stress because relatively lower $H_2O_2$ can trigger LOH in their genomes. Hence, better abilities of mother cells to prevent genotoxic stress factors from leaking to daughter cells also render mothers' own genomes more sensitive to elevated oxidative stress.

## Concerted increase of genomic instability

We observed a large number of multiple LOH events where $H_2O_2$ induced LOH occurred in both mother and daughter cells. We focused on the three-quarter black colonies to illustrate the nature of concerted multiple LOH events. Based on parsimonious reasoning, a three-quarter black LOH can be considered as a joint outcome of a half-black and a quarter-black LOH. The null hypothesis here is that half-black LOH and quarter-black LOH occur independently, which means that the probability of three-quarter black colonies is the product of the probabilities of half- and quarter-blacks, i.e. random expectation. The alternative hypothesis is that events of half-black LOH and quarter-black LOH are associated with and lead to an increased occurrence of three-quarter black LOH. We performed one-tailed Fisher's exact tests on experimental observations with sufficient events of half-, quarter-, and three-quarter black LOH events, and applied Bonferroni correction to adjust for multiple tests. Of the 35 experiments, 21 experiments show that three-quarter LOH events occurred at significantly higher frequencies in all $H_2O_2$ conditions with sufficient number of LOH events (Table S2). In nine experiments, three-quarter LOH events occurred at significantly higher frequencies in all but one $H_2O_2$ condition. Overall, 163 out of 185 measurements show significantly higher three-quarter LOH events than the null hypothesis (Table S2). Hence, the null hypothesis can generally be rejected. We concluded that the high occurrence of three-quarter black colonies is the result of multiple concerted LOH events.
(A)

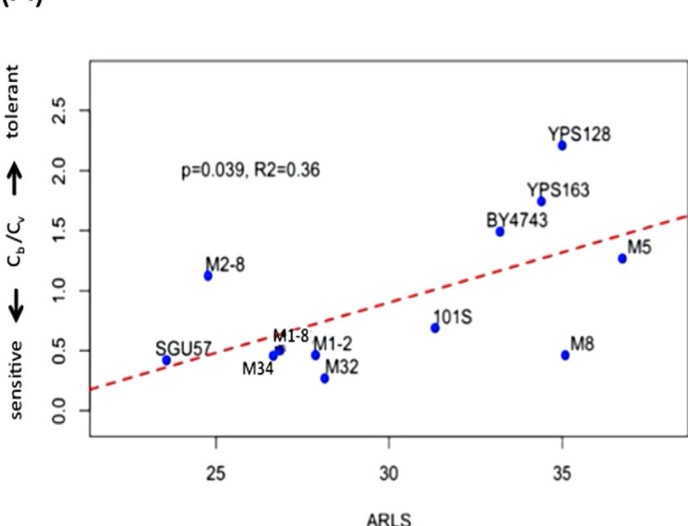

(B)

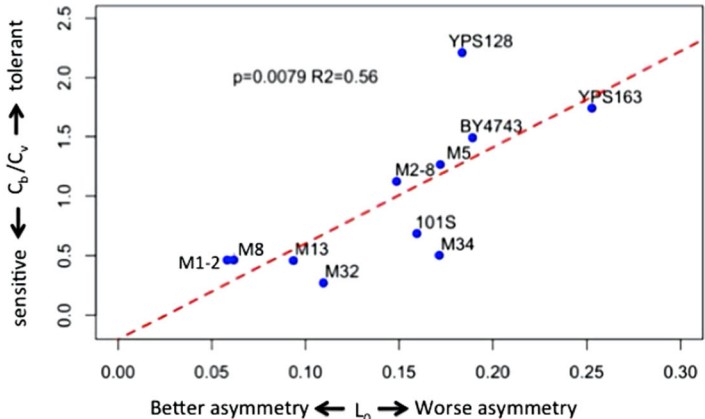

**Figure 3 Hydrogen peroxide induced LOH correlates with replicative lifespan and mitotic asymmetry.** (A) Genomic tolerance to hydrogen peroxide induced LOH, as measured by $C_b/C_v$, correlates with average replicative lifespan (ARLS). (B) Trade-off between tolerance to hydrogen peroxide induced LOH and mitotic asymmetry. In both panels, blue dots represent the average measurements of yeast strains. The red dashed lines are regression lines.

## Contrasting LOH onsets between hydrogen peroxide treatment and chronological aging suggest cells' ability to suppress endogenous oxidative stress during aging

In our previous study on LOH during chronological aging, we found that the onset of LOH often lagged behind the drop of viability. We used $T_g$ to represent the mid-point for the increase of LOH, and $T_c$ as the midpoint for the drop of viability during chronological aging (Fig. 4) (*Qin, Lu & Goldfarb, 2008*). $T_g$ values are often larger than $T_c$ values in the collection of yeast strains that we studied (Fig. 4) (*Qin, Lu & Goldfarb, 2008*).

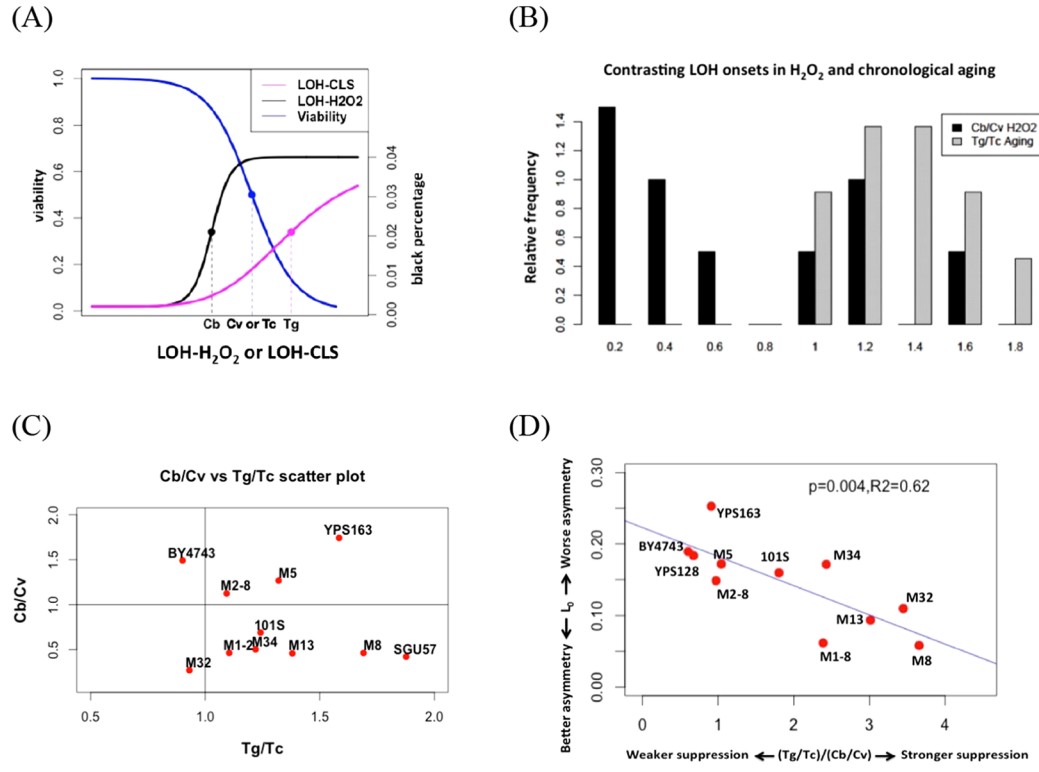

**Figure 4 Contrasting onsets of LOH events between exogenous hydrogen peroxide treatment and chronological aging and its implications.** (A) The mid-transition points of LOH frequencies in exogenous hydrogen peroxide treatment and chronological aging. To illustrate the contrast, we used the same viability curve for both exogenous hydrogen peroxide treatment and chronological aging. (B) Bar plots of $C_b/C_v$ and $T_g/T_c$ ratios. (C) Scatter plots of $T_g/T_c$ and $C_b/C_v$ ratios. (D) Strong negative correlation between mitotic asymmetry $L_0$ and $(T_g/T_c)/(C_b/C_v)$.

In this study, onset of exogenous $H_2O_2$ induced LOH often occurred earlier than the drop of viability, as $C_b$ is often smaller than $C_v$ (Fig. 4). For most natural isolates, this $C_b/C_v$ ratio is lower than 1.0 (Fig. 4). If we assume that a similar level of oxidative stress corresponds to the 50% viability in each strain, this contrast suggests that during chronological aging, the genotoxic effect of endogenous oxidative stress is shielded even when viability has dropped during chronological aging. Hence, the contrast between LOH onsets is a proxy for the extent to how much cells can suppress the genotoxic effect of endogenous oxidative stress on genome integrity. Based on this reasoning, the ratio of $\frac{T_g/T_c}{C_b/C_v}$ indicates the ability of yeast cells to suppress the level of endogenous oxidative stress during the normal chronological aging process. Remarkably, we found a trade-off between mitotic asymmetry and ability of cells to suppress endogenous oxidative stress during chronological aging, as shown by a negative linear correlation between mitotic asymmetry $L_0$ and $\frac{T_g/T_c}{C_b/C_v}$ (Fig. 4D). This finding corroborates the tradeoff observed in Fig. 3B.

## SUMMARY AND DISCUSSION

Our results show that exogenous $H_2O_2$ tends to induce early-onset of LOH increase, in contrast to LOH events caused by endogenous oxidative stress during chronological aging.

We found that the tolerance of yeast strains to exogenous $H_2O_2$ correlated positively with their average RLS and trade-off with mitotic asymmetry. We argue that the contrasting onsets of LOH caused by exogenous and endogenous oxidative stress indicate the ability of cells to suppress genotoxic stress during chronological aging, and this contrast correlates well with mitotic asymmetry. We concluded that a yeast strain's ability to counter genotoxic stress is a good predictor of its RLS and its mitotic asymmetry, although the connection between RLS and mitotic asymmetry is more complicated (Fig. 5A). We presented a plausible explanation to account for the trade-off between genomic tolerance to oxidative stress and mitotic symmetry (Figs. 5B and 5C). Basically, when mother cells have a stronger ability to prevent oxidative damages from leaking to the daughter cells, mother cells put their own nuclear genomes at a higher risk for oxidative damage (Fig. 5B). When mother cells tend to leak more oxidative damages to daughter cells, mother cells have a better chance to protect the integrity of their genomes (Fig. 5C).

Strong evidence supports the role of ROS in yeast replicative aging and asymmetric age partition between mother and daughter cells. Replicatively older cells showed higher levels of superoxide anion and $H_2O_2$ than younger cells, as measured by dihydroethidium and dihydrorhodamine, respectively (Lam et al., 2011; Scheckhuber et al., 2007). Mitochondria in mother cells were found to gradually lose their inner membrane potential, lead to an iron-sulfur cluster defect and nuclear genome instability (Veatch et al., 2009). This mitochondria dysfunction can lead to increased levels of ROS (Breitenbach et al., 2014). Mother cells tend to retain mitochondria with higher superoxide levels than those in daughter cells (McFaline-Figueroa et al., 2011). Consistent with the role of ROS, supplementation of spermine/spermidine, free-radical scavengers, can increase yeast RLS (Eisenberg et al., 2009). We speculate that exogenous hydrogen peroxide most likely disrupts the homeostasis of endogenous ROS and might in turn lead to high levels of superoxide anions. Clearly, further studies are needed to verify these conjectures.

We would like to emphasize that our main conclusions in Fig. 5 should be viewed as a statistical generalization at the population level and are an indirect observation of the role of ROS in yeast aging. Yeast lifespan is a known pleiotropic trait with a large range of phenotypic plasticity. In 'wildtype' yeast strains, the average RLS can vary by more than 2-fold, and the average CLS can change by nearly 5-fold (Qin, Lu & Goldfarb, 2008). The detailed pictures for the role of ROS in yeast aging are fairly complicated at molecular levels (Barros et al., 2004; Lin et al., 2002; Mesquita et al., 2010; Reverter-Branchat et al., 2004; Weinberger et al., 2010), and part of the complexity can be attributed to ecological niches and genetic histories of yeast strains. Recently, several quantitative models related to yeast aging have been developed, though much work is needed to improve these models (Gillespie et al., 2004; Kowald & Kirkwood, 1994; Rashidi, Kirkwood & Shanley, 2012; Sozou & Kirkwood, 2001; Tan et al., 2009; Tang & Liu, 2010). By looking at the statistical trends at the population level, we show a tradeoff between oxidative stress response and mitotic asymmetry. This tradeoff would be difficult to discern at the

(A)

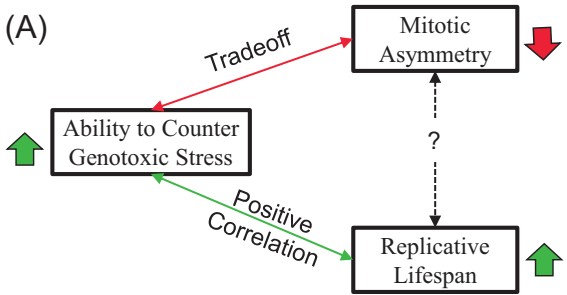

(B)

A yeast strain with good mitotic asymmetry
tends to be sensitive to genotoxic stress.

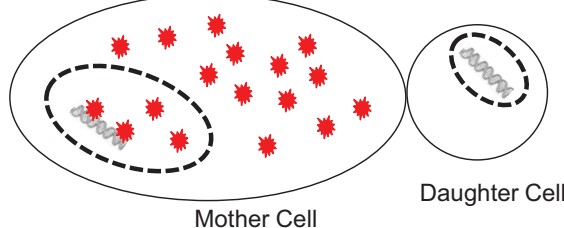

(C)

A yeast strain with poor mitotic asymmetry
tends to be more tolerant to genotoxic stress.

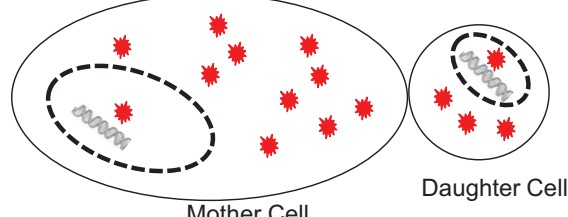

**Figure 5 Main conclusions and a plausible model.** (A) Ability to counter genotoxic stress correlates positively with replicative lifespan, but trades-off for mitotic asymmetry. Consequently, the connection between mitotic asymmetry and replicative lifespan is more complicated. (B and C) A plausible model to account for the trade-off between mitotic asymmetry and tolerance to oxidative stress-induced genomic instability. Red stars represent ROS-induced damages. Mother cells tend to prevent ROS-induced damages to themselves, but this sacrifice put their own chromosomal DNA at an increased risk of ROS induced damages. A cell with better mitotic asymmetry is presented in (B) and a cell with poor mitotic asymmetry in (C). Daughter cells have fewer damages to start with in (B) but more damages in (C). Mother cells are at a high risk for LOH in (B) but low risk for LOH in (C).

molecular level in individual strains. We hope our findings (Fig. 5) will be useful for developing improved mathematical models on yeast aging.

The *MET15* locus is located adjacent to the rDNA locus on chromosome 12, as we discussed previously (*Qin, Lu & Goldfarb, 2008*). Accumulation of extrachromosomal rDNA was shown to cause replicative aging (*Sinclair & Guarente, 1997*). Recently, it

was argued that stability of the rDNA locus is the main determinant of yeast lifespan (*Ganley et al., 2009*; *Kobayashi, 2014*), and rDNA locus stability can be influenced by more than 10% of yeast genes (*Saka et al., 2016*). Increased genomic instability in aging mother cells correlates with increased insertions of retrotransposons at the rDNA locus (*Patterson et al., 2015*). Further studies are needed to investigate the link between oxidative response and rDNA locus, perhaps through next-generation sequencing of multiple yeast strains.

Based on the observed distribution of $C_b/C_v$ in the 12 strains studied (Fig. 4B), there may be two groups of yeast strains: One group with $C_b/C_v$ greater than 1, and one group with $C_b/C_v$ less than 1. We argue that more yeast strains need to be surveyed in order to address this question.

Overall, we conclude that a general picture of yeast aging at the population level can help us better understand how longevity is maintained in populations. With the availability of many yeast strains with sequenced genomes (*Kaya et al., 2015*), *Saccharomyces cerevisiae* has become a promising model organism to understand cellular aging through the integration of molecular biology, population biology, and systems biology.

### Funding
This project was partially supported by the National Science Foundation Award #1022294 and #1453078, an NCMHD grant (NIH 5P20MD000215-05) given to the Spelman Center for Health Disparities Research and Education. The funders had no role in study design, data collection and analysis, decision to publish, or preparation of the manuscript.

### Grant Disclosures
The following grant information was disclosed by the authors:
National Science Foundation Award: #1022294 and #1453078.
NCMHD: NIH 5P20MD000215-05.

### Competing Interests
The authors declare that they have no competing interests.

### Author Contributions
- Emine Guven analyzed the data, contributed reagents/materials/analysis tools, wrote the paper, prepared figures and/or tables, reviewed drafts of the paper.
- Lindsay A. Parnell performed the experiments, analyzed the data, contributed reagents/materials/analysis tools, wrote the paper, prepared figures and/or tables, reviewed drafts of the paper.
- Erin D. Jackson performed the experiments, contributed reagents/materials/analysis tools, reviewed drafts of the paper.
- Meighan C. Parker performed the experiments, contributed reagents/materials/analysis tools, reviewed drafts of the paper.

- Nilin Gupta performed the experiments, contributed reagents/materials/analysis tools, reviewed drafts of the paper.
- Jenny Rodrigues performed the experiments, contributed reagents/materials/analysis tools, reviewed drafts of the paper.
- Hong Qin conceived and designed the experiments, performed the experiments, analyzed the data, contributed reagents/materials/analysis tools, wrote the paper, prepared figures and/or tables, reviewed drafts of the paper.

### Data Deposition

GitHub: https://github.com/hongqin/H2O2LOH_PeerJ2016.

### Supplemental Information

Supplemental information for this article can be found online at http://dx.doi.org/10.7717/peerj.2671#supplemental-information.

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
