# Peer review of "Hydrogen peroxide induced loss of heterozygosity correlates with replicative lifespan and mitotic asymmetry in Saccharomyces cerevisiae"

_PeerJ, doi:10.7717/peerj.2671_

## Round 0.1 · original submission · Minor Revisions

Please revise your manuscript according to the reviewers' comments.

·

Basic reporting

The manuscript is generally well written and appropriately formatted, but there are a few minor grammatical issues that should be addressed. The introduction discusses a number of relevant observations and topics, but a couple additions should be made as noted below.

Introduction- It would be worthwhile to point out the observation in one of the referenced articles (Mesquita et al., 2010) that low doses of hydrogen peroxide can have a hormetic effect on lifespan (promote longevity). Also, it would be helpful to discuss what has been observed regarding changes in ROS levels during replicative or chronological aging, since later in the paper there is discussion of the Tg value reflecting age-dependent oxidative damage to the genome, but no direct measurements of age-dependent ROS are being made. In line 65, note that chronological aging includes aging of daughter cells as well, since many cells in stationary phase cultures are newborn daughters. An additional reference for genome instability during chronological aging would also be worthwhile to more fully represent what has been observed for this topic.

Results- The initial portion of the results section includes the formulas used for calculations of Cb and Cv and a description of the terms, but this discussion seems more appropriate for the methods section.

Figures- The units (molarity, %) for hydrogen peroxide concentration in Figure 2 (and the supplemental figures) should be made clear. The y-axis of Figure 4B should be clearly explained. The graph appears to show the frequency of strains having a given Cb/Cv or Tg/Tc ratio, but it is not clear exactly what the y-axis number represents, since it is not in whole numbers (not simply number of strains in a given ratio bin). Figure 5B would be improved by showing the two relevant scenarios – a mother cell not transmitting damage in contrast to a mother transmitting damage to a daughter.

Discussion- The role of extrachromosomal rDNA circles as a cause of aging is noted, but it would be good to include a broader perspective on the potential role of rDNA circles in aging from one or more recent articles, since they do not appear relevant for all changes in replicative lifespan.

Minor grammatical issues:
Line 71 – “oxidative damages” is a little awkward to read. Oxidatively damaged molecules or oxidative damage would be better.
Line 80 – “and an measure” change an to a.
Line 85 – “genomics instability” change to genomic.
Line 88 – Gene symbol should be in all uppercase italics.
Lines 126-127 – “when MET15+/- is converted into homozygous MET15-/-“ would read better with “cells” or “strains” after MET15.
Line 207 – “Base on this” to “Based on this”
Line 217 – “endogenous oxidative stressed during” change stressed to stress or stresses.
Lines 233+234 – Change folds to fold.
Line 239 – “though much work are need to improve” change to “is needed” or something similar.

Experimental design

The authors clearly define their major goal as an attempt to determine if there is a link between variation in lifespan and oxidative stress-induced genomic instability. There are still many open questions concerning the role of oxidative stress and damage in aging, so the question is meaningful.

Lifespan and mitotic asymmetry values were previously reported and are compared to the new data for viability and loss of heterozygosity in response to varying hydrogen peroxide concentrations.

Specific comments
1- The level of replication was unclear from the methods. The authors state that plates were spread in triplicate for the LOH assays, but was a single population of cells spread between three plates for each trial for each strain, or were replicate populations for each concentration of hydrogen peroxide used for each strain for each trial?

2- Some plates were left out for more than two days for black color development, but would results have changed for other trials/samples if plates were also incubated longer (not clear why a longer incubation was needed in some cases)?

3- The authors state that all strains were assayed multiple times, but stating the range of the number of replicate experiments is preferred so that the level of replication is clear.

4- How viability was measured after treatment with hydrogen peroxide should be explicitly stated. Were colony-forming units for each concentration compared to those for mock or untreated samples, or was viability directly measured using a live/dead cell stain?

Validity of the findings

The main findings discussed are that 1) Cb (LOH in response to hydrogen peroxide) is generally lower than Cv (loss of viability in response to hydrogen peroxide); 2) the Cb/Cv ratio correlates positively with RLS; 3) the Cb/Cv ratio correlates with worse mitotic asymmetry; and 4) the (Tg/Tc)/(Cb/Cv) ratio is inversely correlated with mitotic asymmetry. The data generally appear sound and statistically significant differences are reported to support these observations.

For finding 1, Table S1 shows that six of the 12 strains have Cb values lower than their Cv values (first two columns), and in the ratio column only seven of the 12 strains have a Cb/Cv less than 1. This makes it seem like there may be two different categories of responses, rather than a typical response yielding a Cb/Cv less than one. This variation should be discussed in more detail.

Finding 4 above is interpreted as suppression of endogenous oxidative damage to DNA during chronological aging being inversely correlated with mitotic asymmetry. However, only hydrogen peroxide is used in the current study and the previous work to measure Tg did not directly investigate ROS in general or any particular ROS as directly contributing to Tg. Therefore, it is not completely clear how much Tg depends on endogenous ROS. The authors should therefore do at least one of the following: 1) be more conservative in their interpretation and note the uncertainty of whether Tg is strongly dependent on ROS, 2) review the literature regarding ROS during yeast aging to demonstrate and note whether hydrogen peroxide is likely to be a significant cause of DNA damage in normal aging, 3) test other types of ROS or ROS-inducing agents with their assay system to at least show that the response they see is general to multiple forms of ROS.

Reviewer 2 ·

Basic reporting

The authors have tried to investigate the role of oxidative stress in cellular aging by treating a collection of yeast strains with hydrogen peroxide and examining its effect on genome instability and mitotic asymmetry. In fact this study is a short extension of their previous study published in 2008.

Major comments:
1. They should explain how they measured ARLS more clearly. It is not clear how they assayed RLS.
2. What were the concentrations of H2O2 used in this study? It has not been specified.
3. In page 8 line 148, The authors have stated that "we detected a strong association between Cv and Cb with R^2 = 0.80". They should provide a plot for the same and provide the Cb and Cv values in supplementary table.
4. Pairwise T-test cannot be used for measuring non-random association, the authors should use Fishers Exact test to test if the three quarter blacks are significantly higher than random expectation in page 9, line 187.
5. There are several typos and grammatical errors (mixing passive voice and active voice in a sentence, using an in the place of a) in the paper. The authors are suggested to check them carefully.
6. Line 31 of the abstract is not clear, it should be rephrased.

Experimental design

No comments

Validity of the findings

Since, mechanism underlying the observed relationship between oxidative stress, LOH, mitotic asymmetry and RLS has not been dealt with in this study, the associations detected in this paper is indirect and should be clearly stated in the paper. The authors are welcome discuss or speculate plausible mechanisms underlying H2O2 can affect RLS in more detail. This would make the paper more interesting.

---

## Round 0.2 · accepted · Accept

The authors have answered the questions/comments of the reviewers, so I suggest to accept it.

Reviewer 2 ·

Basic reporting

I am satisfied by their response and the present form of the manuscript

Experimental design

No further comments on their design

Validity of the findings

No comments on their findings